# Exploring Physical Activity Levels, Barriers, and Education Sources in People with Cancer Undergoing Chemotherapy

**DOI:** 10.3390/cancers17182987

**Published:** 2025-09-12

**Authors:** Rebecca Cesnik, Kellie Toohey, Nicole Freene, Stuart Semple

**Affiliations:** 1ACT Government, Health and Community Services Directorate, Canberra 2601 (ACT), Australia; 2Faculty of Health, University of Canberra, Canberra 2617 (ACT), Australia; kellie.toohey@scu.edu.au (K.T.); nicole.freene@canberra.edu.au (N.F.); stuart.semple@canberra.edu.au (S.S.); 3Research Institute of Sport and Exercise, University of Canberra, Canberra 2617 (ACT), Australia; 4Physical Activity, Sport and Exercise Research Theme, Faculty of Health, Southern Cross University, Gold Coast 4225, Australia; 5Health Research Institute, University of Canberra, Canberra 2617 (ACT), Australia

**Keywords:** exercise, physical activity, cancer, barriers, education

## Abstract

Physical activity during chemotherapy is important to improve side effects, survival rates, and treatment completion, yet most people do not reach physical activity guidelines. This study aimed to understand the physical activity levels, barriers, and education sources within the ACT, Australia. Understanding the reported barriers to physical activity—including side effects, motivation and time—and the desire for education offers opportunities to address gaps in services and ensure physical activity is integrated into chemotherapy care.

## 1. Introduction

Cancer is a major cause of mortality and morbidity globally, with 14 million new cases diagnosed each year [1]. Chemotherapy is a systemic cancer treatment with curative, life-extending, or palliative intent [2] that has significant negative physical and psychological side effects. There is compelling evidence that engaging in physical activity (PA) and exercise during chemotherapy may improve physical function [3], strength [4], pain [4], quality of life [3], aerobic capacity [3], fatigue [5], depression [3], chemotherapy-induced peripheral neuropathy [6], and long-term improvements in cognitive fatigue, cardiorespiratory fitness, and strength [7]. PA may also improve cancer outcomes [8], chemotherapy tolerance and completion [9], hospital admission rates, the length of stay [10], and survival rates [11], among other benefits.

Physical activity is defined as “any bodily movement…that requires energy expenditure”, whereas exercise is a subset of PA that is planned and repetitive [12]. The optimal dose of PA for people undergoing chemotherapy is unknown. However, recommendations align with World Health Organisation (WHO) guidelines; that is, a minimum of 150 min of moderate-intensity aerobic activity or 75 min of vigorous-intensity activity per week (or a combination of the two); and two to three sessions per week of moderate-vigorous resistance exercise [12,13,14,15]. Not all individuals will meet these guidelines at all points in the cancer continuum. For these individuals, targeted exercise prescription and avoiding inactivity is recommended, as smaller doses of PA is sufficient to achieve some physical and psychological benefits [14,15].

While the substantial benefits of PA during chemotherapy are well-documented—including improved physical and psychological outcomes—there remains a persistent and significant shortfall in PA participation among individuals undergoing chemotherapy [16]. Chemotherapy itself is consistently identified as one of the most significant predictors of inactivity [17] compounding existing health challenges and diminishing patient outcomes. Numerous barriers, such as distressing treatment side effects, inadequate knowledge or guidance, limited encouragement from social networks and healthcare professionals, and competing life commitments and hospital appointments, consistently hinder engagement in PA [18,19,20]. Without effective interventions, PA levels in this population not only stagnate but often decline [16], risking increased morbidity and reduced quality of life during a critical phase of cancer treatment. Despite increasing recognition of PA’s benefits, there is a critical gap in the literature: to our knowledge, no studies to date have specifically examined PA levels, barriers, or education sources solely among patients undergoing chemotherapy within the Australian context. As healthcare systems strive to deliver evidence-based supportive care, this knowledge gap limits the development of targeted interventions and policies that could support patients during chemotherapy and optimise health outcomes.

Therefore, the objective of this study was designed to fill this crucial void. Its primary aim is to assess self-reported PA levels in people undergoing chemotherapy, classifying respondents as ‘active’ or ‘insufficiently active’ according to established guidelines [12,13,14,15]. The secondary objectives were to identify the self-reported barriers preventing PA and to evaluate the nature and sources of PA education available to patients during chemotherapy. By addressing these objectives, the study seeks to inform the development of integrated, evidence-based PA programs and policies that could profoundly enhance patient care and well-being during chemotherapy.

## 2. Materials and Methods

### 2.1. Study Design

A cross-sectional mixed methods questionnaire was distributed at all outpatient chemotherapy sites within Canberra, Australian Capital Territory (ACT), Australia over four weeks to capture various chemotherapy cycle lengths (November to December 2018). This time period ensured that most people attending the service had the opportunity to be approached while reducing the risk of duplicate invitations, which has implications for data integrity and participant and staff fatigue. Respondents were sourced from all chemotherapy centres across the region, which incorporates a population size of over 450,000 [21], public and private services, and people living in metropolitan and regional areas. Participants were included in the study if they were 18 years or older and currently undergoing outpatient chemotherapy for any type or stage of cancer. Participants were excluded if they were unable to read or write in English or had cognitive disfunctions that would limit their ability to respond to the questionnaire.

The participants were invited to participate by independent nursing staff who were not involved in the study. Staff were requested not to encourage people to partake if they declined, but to document reasons for not participating. Potential participants were advised that involvement was voluntary and that staff had no connection to the study to reduce response bias. A Participant Information Sheet was provided, and the completion of the attached questionnaire followed this implied consent. Questionnaires were completed during chemotherapy sessions and placed in a sealed box by nurses.

Participants self-reported their cancer type, which was subsequently classified using ICD-10-CM codes [22] to enable consistent and streamlined reporting across the study. The study design was approved on 26 September 2018 by the ACT Health Human Research Ethics Committee (2018/LRE/177).

### 2.2. Sample Size

The sample size required to accurately represent the eligible patients attending the chemotherapy sites during the study period (*n* = 459) was determined using an online survey sample size calculator. Based on a 95% confidence level and 5% margin of error, 209 respondents were required [23].

### 2.3. Survey

The study utilised a validated questionnaire [24] and study-specific questions. Study-specific questions were used to identify demographics, cancer type, stage, treatment location, chemotherapy commencement date, barriers to PA, and sources of PA education. Cancer types were grouped by ICD-10-CM code. Likert scales were used to identify the level of agreement about patients’ experiences of education and capacity. Participants were asked to select barriers and sources of PA education that applied to them via multiple-choice questions. Survey options were developed based on the current literature [19,25,26,27]. Qualitative responses were obtained through optional opportunities to further elaborate on multiple choice questions. The questionnaire was validated for face and content validity before use [28]. Ten people undergoing chemotherapy and six experts were asked to qualitatively assess face validity, making comments about clarity, understandability and relatedness [28]. Slight adjustments were made to questions as they suggested, but no questions were omitted. The adjusted questions were provided to eight experienced oncology clinicians for content validity. Respondents were asked to rate each question from 1–4 for covering all dimensions, clarity, and relevance [28]. The content validity index was calculated at the item-level (I-CVI) and scale-level (S-CVI). If I-CVI was higher than 0.79, it was considered acceptable; if the score was 0.70–0.79, it needed revision, and less than 0.70 was excluded [28]. All items were deemed acceptable (I-CVI = 0.9–1.0; S-CVI = 0.9) (Appendix A).

Physical activity levels were assessed using a modified Godin–Shephard Leisure Time Questionnaire (GSLTQ) [29]. The GSLTQ is a 3-item self-administered questionnaire, that asks for the frequency of mild, moderate, and strenuous LTPA bouts of at least 15 min duration in a typical 7-day period [29]. It is commonly used in cancer populations and has been validated to classify participants as ‘active’ or ‘insufficiently active’ using leisure time PA and the leisure score index (LSI) [LSI = (frequency of moderate leisure time PA × 5) + (frequency of strenuous leisure PA × 9)]. People with a LSI ≥ 24 were classified as ‘active’, while those who scored ≤23 were classified as ‘insufficiently active’ [24]. The GSLTQ was modified to ask for the average duration of the activity to calculate the average PA per week. This is the most common modification of the GSLTQ used in oncology settings, used in over 70% of studies [30]. Average PA was compared to WHO moderate-vigorous PA (MVPA) guidelines [moderate intensity + 2 × vigorous intensity] [31].

In summary, this section describes a comprehensive approach to capturing both quantitative and qualitative aspects of PA among patients undergoing outpatient chemotherapy across a diverse regional sample. Aerobic PA levels were measured using the modified GSLTQ, collecting data on frequency (times per week) and average time per session, with frequency serving as the primary variable in subsequent analyses according to established GSLTQ methodology. Average PA minutes per week were also calculated to facilitate comparison with WHO guidelines. Resistance exercise was assessed by asking participants to report frequency (times per week), intensity, and the location of their activities, providing a nuanced understanding of resistance training behaviours within the cohort. This mixed-methods approach allowed for the robust comparison of PA patterns and associated variables, contributing valuable insight into the exercise behaviours of individuals receiving chemotherapy.

### 2.4. Data Analysis

Descriptive analyses were completed to summarise survey data. Barriers and education sources were categorised as ‘yes’/‘no’ for each variable to allow for binary categorical analysis. Associations between resistance exercise or PA levels and variables were calculated using Pearson’s chi-squared tests of independence. Fisher’s exact test was used where cells had counts of less than five. A series of univariable binary logistic regression analyses were completed to identify possible relationships between variables and PA or resistance exercise categories. Bootstrapping analysis, with a repetition of 1000, was used to reduce variability given the non-normal distribution and small sample size. Firth’s logistic regression was used for predictors with quasi-separation. Where appropriate, statistical significance was set at *p* < 0.05 with data presented as mean (M) ± standard deviation (SD). Quantitative analysis was completed using Statistical Package for Social Sciences (SPSS) version 29.0.0.0 (RRID:SCR_002865) and Stata/MP 18.0 (RRID:SCR_012763).

Qualitative data were collated in Microsoft Excel and inductively themed [32] by the primary researcher in consultation with all listed authors.

The Strengthening the Reporting of Observational Studies in Epidemiology (STROBE) reporting guidelines were followed to ensure transparency and completeness in the presentation of methods and results throughout the study [33].

## 3. Results

A total of 111 surveys were collected during the study period. Participant demographics and survey responses are presented in Table 1. The average age of respondents was 59.2 years (range 18–91), with a greater proportion of males than females, and more not working versus working individuals. The most common cancer diagnoses were breast or digestive system cancers. Information about why people chose not to participate was not routinely collected, limiting our ability to identify specific reasons for non-participation.

The GSLTQ classified 11.7% (*n* = 13) of respondents as ‘active’ and 87.4% (*n* = 97) as ‘insufficiently active’. Similarly, using average PA from the modified GSLTQ, only 9.9% (*n* = 11) met WHO MVPA aerobic guidelines. There were 79 participants (72.5%) who reported no leisure time MVPA. On average, respondents participated in 9.64 min (SD ± 37.51) of vigorous activity, 35.81 min (SD ± 83.26) of moderate activity, and 81.50 min (SD ± 131.76) of mild activity in the past seven days.

Participants completed an average of 0.9 (SD ± 1.74) resistance exercise sessions in the last week. Twenty-three respondents (20.7%) reported completing 2–3 sessions; however only 9.9% (*n* = 11) reported moderate-vigorous intensity. Participants mostly completed resistance exercise at home (64.5%), with 16.1% reporting attending a gym and 16.1% attending a class. Overall, 78.7% (*n* = 85) reported they felt their PA levels had decreased whilst being on chemotherapy (13 did not respond, and 17 reported no decline).

### 3.1. Barriers to Physical Activity

Fatigue was the most frequently reported barrier, (72.1%, *n* = 80), followed by strength (35.1%, *n* = 39), pain (30.6%, *n* = 34), motivation (26.1%, *n* = 29), nausea (21.6%, *n* = 24), and a lack of time (8.7%, *n* = 9) (Table 2). Decreases in strength, energy, and ability to complete daily activities were reported by 92.6% of respondents. Reporting strength as a barrier significantly decreased the likelihood of being classified as ‘active’ compared to no barrier (Odds Ratio (OR) = 0.05, 95%CI 0.00, 0.95 *p* < 0.05), with no respondents who identified strength as a barrier being classified as ‘active’. The barrier of time significantly increased the likelihood of participating in resistance exercise compared to no time barrier (OR = 6.29, 95% CI: 1.33, 29.53, *p* < 0.05). No other barriers were statistically related to GSLTQ activity classification or resistance exercise.

Qualitative responses identified other barriers to PA, including peripheral neuropathy, the progression of disease, the cyclic nature of symptoms, and preferring social support when exercising (Table 3). Others discussed their history of PA supporting their ability to maintain PA during chemotherapy. “I have always been a fit/strong person who believed in strenuous exercise prior to treatment, which made it easy to continue” (Female, 47, cancer type not reported).

### 3.2. Education Sources

Nurses were the most common source of PA education (*n* = 34). Other frequent sources included specialists (*n* = 22), GPs (*n* = 18), and exercise physiologists (AEP) (*n* = 14). Many respondents (*n* = 34, 32.4%) reported receiving no PA education during chemotherapy from any source (Table 4).

While 63.0% (*n* = 70) reported they agreed or strongly agreed that they received sufficient PA education, 48.7% (*n* = 54) stated they wanted more. Only 16.2% (*n* = 17) of participants identified being referred to an AEP or physiotherapist at any time through their treatment. Identifying AEPs as a source of education significantly increased the likelihood of participating in resistance exercise compared to those who did not (OR = 1.43, 95% CI: 0.25, 2.79, *p* < 0.05). No other sources of education were significantly related to GSLTQ activity classification or resistance exercise.

Several patients discussed education in their open-ended responses (Table 3). Some discussed their individual symptoms and need for support: “I am in a unique situation in that I am heavily pregnant whilst having chemotherapy and have been very unwell. I would like to know what exercise I can do and personalised support/specific referral to someone who can assist.”(Female, 33, breast cancer); “I have never been told that I should do exercise during my chemotherapy. If this is good for me and encouraged, I can push myself to do so. Please let me know” (Female, 57, ovarian cancer). On the other hand, others discussed the benefits of being referred into PA services: “I made enquiries about exercise early in my treatment as I was not knowledgeable about what I was suitable to do. Having seen an exercise physiologist, I do feel more confident and know my limits.” (Female, 62, lung cancer). The need for PA services to be integrated into the chemotherapy program was also identified: “I think it would be valuable to integrate formal structured exercise into the chemotherapy program.” (Female, 56, uterine cancer).

## 4. Discussion

This study provides valuable insights into the alarming trend of aerobic PA levels among people undergoing chemotherapy: namely, only 17% of those surveyed in this study were classified as “active”. These findings align with the existing literature, which consistently demonstrate low PA adherence across the cancer continuum [16,34,35]. While inactivity affects approximately one-third of the adult population [12] and cancer survivors [36], this study revealed significantly higher levels of inactivity among individuals undergoing chemotherapy, with 72% reporting no MVPA. Additionally, research in this specific location suggests that 70% of adults do meet PA guidelines [37], indicating a substantial difference to people undergoing chemotherapy within this study. Importantly, higher PA levels have been linked to positive outcomes in maintaining independence, daily activities, personal care, and mobility [34]. This is particularly relevant given that a large proportion of respondents in our study reported a decline in these outcomes. The established benefits of meeting aerobic PA guidelines for people with cancer have significant impacts to individuals and the health system, including reduced hospital admissions and length of stay [10], increased chemotherapy adherence [9], and increased survival rates [8]. While aerobic PA after chemotherapy treatment may improve aerobic capacity, many patients will experience the lasting impacts of this deterioration. Addressing barriers and promoting access to PA education through targeted interventions and adapting services should be a priority for healthcare organisations and practitioners.

This study found that over 70% of respondents participated in no resistance exercise. Resistance exercise consistently results in statistically and clinically significant improvements in strength, function, fatigue, and quality of life during and after cancer treatment [15]. Low muscle mass during cancer treatment is a predicator of all-cause and cancer-related mortality and of increased treatment-related side effects [38]. Inactivity exacerbates a negative cycle of strength loss [39], which is supported by the significant relationship between strength and decreased PA found in this study. Chemotherapy can induce muscle loss through various mechanisms, including direct toxicity to muscle fibres, systemic inflammation leading to protein breakdown, hormonal disruptions decreasing anabolic hormones, nutritional deficiencies due to treatment side effects, and inactivity-induced muscle atrophy [39]. This multifactorial nature underscores the importance of a comprehensive approach to prevention and management [39]. Resistance exercise has been shown to improve muscle mass and strength during and after treatment [39], preventing cancer-induced sarcopenia [40]. Tailored resistance exercise programs should be routinely offered throughout the chemotherapy journey to reduce prevalence of this significant side effect.

The side effects of cancer and chemotherapy treatment were the greatest reported barrier to PA in this study. This is consistent with other studies for people undergoing chemotherapy [25,35,41,42]. A higher incidence of side effects can result in reduced quality of life [43]. PA is an effective treatment for many short- and long-term side effects [14,15], and reduced side effects have been demonstrated in those who are physically active [44]. Given the significant impact side effects have, trained professionals need to provide individualised support to help people undergoing chemotherapy to remain active and manage their treatment-related side effects. To better support people undergoing chemotherapy to be active, individualised support should include tailored exercise programs, ongoing guidance from trained professionals, and clear, accessible education about the benefits of and strategies for PA. Current data show that addressing barriers like fatigue, time constraints, and gaps in professional knowledge while ensuring easy access to specialist referrals and practical support are essential to help individuals maintain or increase their activity levels during treatment.

Cancer-related fatigue was the greatest singular barrier in this study and is widely recognised as the most prevalent and debilitating symptom experienced by individuals with cancer [5]. It has been repeatedly identified as the largest single barrier to PA in people undergoing chemotherapy [26,27,45]. The heightened severity of cancer-related fatigue is reported among those undergoing chemotherapy compared to other treatment types [5], leading to decreased concentration, motivation, and an increased risk of falling [46]. The most effective treatment for cancer-related fatigue is PA [5,14]. Additionally, physical inactivity is likely to exacerbate fatigue, resulting in greater symptoms [14]. To optimise participation in PA and support fatigue management, services that are tailored to specific needs and fatigue levels need to be accessible among people undergoing chemotherapy. Some strategies that could be effective include embedding accredited exercise professionals, such as AEPs or physiotherapists with cancer expertise, within oncology care teams to provide tailored assessments and supervised exercise prescriptions. Implementing structured, evidence-based exercise programs—whether in-person or via telehealth—has been shown to improve adherence, safety, and outcomes for people with cancer. Additionally, regular staff education sessions and developing clear referral pathways can empower clinicians less familiar with exercise to confidently initiate PA discussions and connect patients to appropriate services. Providing written resources, group-based support, and leveraging technology for remote monitoring and encouragement may further help patients overcome barriers and maintain activity throughout treatment.

Whilst time and other commitments are commonly reported as the main barriers to PA in adults worldwide [47,48], individuals undergoing chemotherapy may require specific support to overcome these barriers. In our study, time was the third highest reported barrier after the side effects of cancer or treatment and motivation. Research demonstrates that time constraints, service availability, cost and feasibility of transport [49], and changing priorities [19] are barriers to PA whilst undergoing chemotherapy. Particularly for those who have limited history with PA or are unable to participate in their previous activities, personalised support to be active in a meaningful way may be needed [50]. Research suggests that people undergoing chemotherapy would like tailored professional support, ensuring they feel safe and comfortable, and can develop strategies to prioritise PA [51]. The qualitative responses of this study support this finding, with those surveyed indicating a need for individualised support or benefits from the individualised support they had received. Changing behaviour is complex and requires the support of trained professionals to facilitate PA integration based on individual goals and needs [52].

In this study, over 60% of participants reported that they were provided PA education from one or more sources. Despite this, almost half reported they would like more information about PA. This suggests that the current sources of education are ineffective to meet participants’ needs. This trend is similar to other studies of cancer survivors [53], indicating individuals’ desire for more information. A recent study in people undergoing chemotherapy indicated similar trends with education, whereby 73% of participants received guidance about PA during treatment [35]. Evidence suggests that individuals are more likely to agree that they can be physically active when they are educated about PA during chemotherapy [27]. Nurses, medical specialists, and GPs were the most common sources of education in this study. While they have the most regular contact with people undergoing chemotherapy, they may not have the expertise or capacity to provide detailed education and advice about PA [54]. While advice from oncologists may influence uptake in PA [52,55], a large proportion of health care professionals self-report having insufficient knowledge of PA during cancer [54,56,57] and often do not initiate PA conversations [56] due to time constraints, lack of knowledge, or safety concerns [57]. Additionally, while referrals to exercise professionals may be the most beneficial action taken by clinicians [55], evidence suggests this may be limited due to lack of services in health centres [18,58], extra workload to screen and refer [59], knowledge of benefits and scope [52], believing it is too hard to participate during treatment [52,58,59], unknown quality of services, and funding arrangements [52,59]. Support must be provided to ensure all health professionals feel confident initiating discussions about PA and providing appropriate advice, screening, and referrals.

The Clinical Oncology Society of Australia (COSA), recommends integrating exercise into standard cancer care, with all health professionals actively promoting PA as a crucial component of treatment [13]. It is essential that advice and referral opportunities are offered regularly [52]. In this study, it was found that, although a range of health professionals discussed PA, this education was inconsistent and rarely followed up with referrals to exercise specialists. The frequency of referrals to exercise specialists was lower in this study than in other studies [56]. Given the desire for more education found in this study, this is a considerable gap in practice that must be addressed if PA is to be incorporated into standard care.

To align with recommendations and ensure optimal care, referral processes to AEPs or physiotherapists with cancer expertise should be streamlined or integrated within the current multidisciplinary team. This will ensure that people undergoing chemotherapy have access to evidence-based information, education, and support to safely engage in PA. In this study, AEPs were the only source of education that statistically increased the likelihood of participating in resistance exercise. PA interventions during treatment are more likely to be cost-effective if they are delivered by AEPs or physiotherapists compared to self-managed PA [60], indicating the importance of simple, consistent referral pathways for both consumers and healthcare organisations. The gaps in PA education in this study highlight the need to review current practices. Specifically, we need to integrate regular education into standard care and ensure that the need for individualised support is effectively identified and routinely offered.

This study provides a valuable insight into the reported PA levels, barriers, and educational sources for people undergoing chemotherapy and was the first to identify PA levels, barriers, and education sources during chemotherapy in Australia. Respondents were sourced from all chemotherapy centres across the region, which incorporates a population size of over 450,000 [21], public and private services, and people living in metropolitan and regional areas, allowing results to be generalised to other regions. While the heterogenous nature of the data can make it difficult to draw significant research conclusions, it is clinically relevant, as it represents real-world chemotherapy settings where a range of cancer types, stages, and treatments are treated simultaneously, and strategies must therefore have broad applicability. This study identifies areas for future research as well as important considerations for clinical services, including an understanding of the small number of participants that are participating in any MVPA, a desire for more PA education, and barriers to PA that health professionals should discuss with this population.

There are some limitations to this study. Firstly, the expected sample size was not reached during the study period. The study was limited to a four-week timeframe, which may have constrained the sample size. This duration was chosen to capture a representative snapshot of patients while minimising duplicate recruitment and participant fatigue. It also reduced the resource burden on staff assisting with data collection, meaning that extending the time period to increase sample size was not appropriate. Despite this, the number of respondents was similar to that seen across the current literature [16]. While PA levels and barriers are similar to what is seen in international research, the limited sample size may impact the generalisability of the research. Secondly, due to the research aims, survey design, and sample size, analysis was limited, and further research is required to understand the relationships between PA levels, barriers, and education sources. Thirdly, all responses were collected from self-reported responses, which could possibly contain bias from recall and health literacy and bias in who chose to be involved. Whilst the GSLTQ is validated in cancer populations, research found that 81% of studies use a modified version for people in oncology research [30]. The most common method of modifying the GSLTQ is to collect the average duration of the activity [30], which was performed within this study. This variation may add recall burden to the respondent [30]. A major limitation of this study is the considerable heterogeneity introduced by its broad inclusion criteria encompassing any cancer type, stage, and all chemotherapy regimens, as there are substantial differences in exercise behaviours, determinants, and treatment toxicities across these variables; however, this diversity also serves as a strength, reflecting the real-world complexity of patients presenting to healthcare settings and underscoring the need for interventions that are effective and adaptable across a wide spectrum of clinical scenarios.

This study identifies several directions for future research. Understanding barriers and facilitators to PA in this population will help tailor effective services. Input from staff, carers, and patients is vital to adapt support to their needs. Future studies should explore the optimal timing, content, delivery, and format of PA education during chemotherapy. Further investigation is needed on how education from AEPs and barriers like time and strength influence participation in MVPA and resistance exercise. Clarifying the factors linking PA, education, and barriers will guide the development of services that maximise PA for people undergoing chemotherapy. Finally, studies with larger sample sizes that aim to identify changes in PA ranging from pre-diagnosis and pre-chemotherapy to post-treatment would be beneficial to gaining a full understanding of the impact that chemotherapy has on PA levels.

## 5. Conclusions

This study reveals a critical gap in PA levels among individuals undergoing chemotherapy in Australia, with a substantial proportion of people engaging in no MVPA or resistance exercise. Alarmingly, nearly one-third of participants in this study reportedly received no PA education during their treatment. This highlights the need for improved services to promote PA and integrate exercise specialists such as AEPs into standard cancer care. All individuals undergoing chemotherapy can benefit from being physically active, and it is essential that they are offered this opportunity. Consistent and reliable individualised PA support is crucial for empowering patients to engage in safe and effective PA during their treatment, ultimately improving their quality of life and overall health outcomes.

## Figures and Tables

**Table 1 cancers-17-02987-t001:** Participant details.

Characteristic	*n* (%)
Age (years)	59.2 (15.8) *
Gender	
Males	72 (64.9)
Females	39 (35.1)
Indigenous (Background)	
Aboriginal	2 (1.8)
Torres Strait Islander	0 (0.0)
Neither	108 (97.3)
No response	1 (0.9)
Education Level	
<Year 10	14 (12.6)
Year 11/12	23 (20.7)
Cert III/IV	19 (17.1)
Diploma	11 (9.9)
Bachelor’s degree	23 (20.7)
Post graduate degree	13 (11.7)
No response	6 (5.4)
Relationship Status	
In a relationship	68 (61.3)
Not in a relationship	38 (34.2)
No response	4 (3.6)
Work Status	
Not working	79 (71.2)
Working hours:	
<10 h/week	1 (0.9)
10–19 h/week	2 (1.8)
20–29 h/week	3 (2.7)
30–35 h/week	6 (5.4)
Full-time (>35 h)	16 (14.4)
No response	4 (3.6)
Cancer Type	
Lip, oral cavity & pharynx	1 (0.9)
Digestive organs	30 (27.0)
Respiratory system and intrathoracic organs	5 (4.5)
Bone	1 (0.9)
Skin	1 (0.9)
Mesothelial and soft tissue	3 (2.7)
Breast	25 (22.5)
Female genital organs	7 (6.3)
Male genital organs	4 (3.6)
Urinary tract	1 (0.9)
Blood and lymphatic system	9 (8.1)
Other	1 (0.9)
No answer	23 (20.7)
Cancer Stage	
Stage 1	4 (3.6)
Stage 2	16 (14.4)
Stage 3	17 (15.3)
Stage 4	32 (28.9)
No response	42 (37.9)
PA levels (GSLTQ Category)	
Insufficiently active	97 (87.4)
Active	13 (11.7)
No response	1
Resistance Exercise	
No resistance exercise	80 (72.1)
Any resistance exercise	31 (27.9)
≥2 times per week at moderate-vigorous intensity	11 (9.9)
Self-reported decrease in activity levels	
Yes	17 (15.3)
No	81 (72.9)
No response	13 (11.7)
Self-reported decrease in strength, energy, or activities of daily living	
Strongly agree	45 (40.5)
Agree	40 (36.0)
Unsure	9 (8.1)
Disagree	8 (7.2)
Strongly disagree	6 (5.4)
No response	3 (2.7)
I have received sufficient education about PA during my treatment	
Strongly agree	15 (13.5)
Agree	55 (49.5)
Unsure	21 (18.9)
Disagree	11 (9.9)
Strongly disagree	7 (6.3)
No response	2 (1.8)
I would like more information about PA during treatment	
Strongly agree	16 (14.4)
Agree	38 (34.2)
Unsure	27 (24.3)
Disagree	24 (21.6)
Strongly disagree	2 (1.8)
No response	4 (3.6)
I have been referred to an exercise specialist during my treatment	
Exercise physiologist	11 (9.9)
Physiotherapist	6 (5.4)
Neither	87 (78.4)
Unsure	5 (4.5)
No response	2 (1.8)

* Mean (±SD).

**Table 2 cancers-17-02987-t002:** Self-reported barriers to PA categorised by PA levels as per GSLTQ and resistance exercise.

Barrier to PA	Total (*n* = 104) *n* (%) ^^^	‘Active’ (*n* = 18) *n* (%) ^^^	Any Resistance Exercise (*n* = 29) *n* (%) ^^^
Fatigue	80 (77.0)	7 (58.3)	22 (75.9)
Strength	39 (37.5)	0 *^a^ (0.0)	9 (31.0)
Pain	34 (32.7)	3 (25.0)	10 (34.5)
Motivation	29 (27.9)	4 (33.3)	7 (24.1)
Nausea	24 (23.1)	0 * (0.0)	5 (17.2)
Lack of time	9 (8.7)	1 (8.3)	2 (6.9)
Unsure what to do	9 (8.7)	3 * (25.0)	6 *^b^ (20.7)
Access to services	7 (6.7)	1 (8.9)	2 (6.9)
Don’t like exercise	5 (4.8)	0 (0.0)	1 (3.4)
Confidence	5 (4.8)	0 (0.0)	2 (6.9)
Told not to	4 (3.8)	0 (0.0)	0 (0.0)
Family worried	3 (2.9)	0 (0.0)	0 (0.0)
Don’t need to	1 (1.0)	0 (0.0)	0 (0.0)
Other	17 (16.3)	3 (25.0)	6 (20.7)

^^^ Participants may have multiple barriers to PA. * Chi-Square *p* < 0.05. ^a^ Odds Ratio = 0.05, 95%CI 0.00, 0.95; *p* < 0.05. ^b^ Odds Ratio = 6.29, 95% CI: 1.33, 29.53; *p* < 0.05.

**Table 3 cancers-17-02987-t003:** Inductive themes.

	Theme	Quote
Barriers	Side effects	“Peripheral neuropathy restricts my hands and feet (feels like they’re covered in scales or feathers)” (Female, 66, bowel and liver cancer)
		“Until recently went to the gym. Due to foot/leg problems had to give up as couldn’t drive. Am considering going to an exercise physiologist” (Female, 72, pancreatic cancer)
	Prioritising other commitments	“I try to be as active as possible. A poor prognosis has me spending more time with my family and less time at a gym” (Male, 57, pancreatic cancer)
	Not told to	“The only guidance on exercise is in the EVIQ patient info sheet which has on p7 “hand foot syndrome—avoid unnecessary walking, jogging or exercise” “ (Male, 65, bowel cancer)
	Lack of social support	“I find it easier to partake in appropriate exercise with someone else—I unfortunately spend most of my time alone” (Female, 70, cervical cancer)
Facilitators	History of PA	“I have always been a fit/strong person who believed in strenuous exercise prior to treatment, which made it easy to continue. I believe I have remained relatively well and strong during treatment due to my exercise ethic.” (Female, 47, cancer type not reported)
	Benefits of PA supports received	“I am part of cancer rehab at UC. This program over 16/52 has been very beneficial. I want to keep going. It has helped my confidence” (Female, 71, breast cancer)
		“I have been attending an ACT Health Program Living a Healthy Life with Chronic Conditions’ and find it has motivated me and given me strategies for exercising and dealing with my condition” (Female, 82, peritoneum cancer)
Preferences	Need for support	“I would like to know as much as possible to strengthen myself and how to manage my exercise to gain the best possible outcome.” (Male, 48, pancreatic cancer)
		“More information on types of exercise from a professional would be useful” (Male, 62, bowel cancer)
	Desire to do more PA	“I realise that I should be doing more and shall do light stuff. Most of my previous has been outdoors land care.” (Male, 78, cancer type not reported)
		“I really miss my daily swim of 1 km and then an aqua aerobic class. Had to curtail this due to increased risk of catching something”. (Female, 69, cancer type not reported).
	Would like integrated PA services	“I think it would be valuable to integrate formal structured exercise into chemo program”. (Female, 56, uterine cancer)
	I do enough PA	“I walk every day and climb 4 flights of stairs twice weekly” (Male, 67, cancer type not reported)
		“No concerns about the lack of resistance exercise at the moment as I have enough day to day (domestic) activities” (Male, 79, neuroendocrine cancer)

**Table 4 cancers-17-02987-t004:** Sources of PA education categorised by PA levels as per GSLTQ and resistance exercise.

Sources of Education	Total (*n* = 105) *n* (%) ^^^	‘Active’ (*n* = 13) *n* (%) ^^^	Any Resistance Exercise (*n* = 31) *n* (%) ^^^
Exercise physiologist	14 (13.3)	1 (7.7)	8 *^c^ (25.8)
Family member	7 (6.7)	2 (15.4)	2 (6.5)
Friend	7 (6.7)	0 (0.0)	1 (3.2)
GP	18 (17.1)	2 (15.4)	3 (9.7)
Independent research	10 (9.5)	1 (7.7)	3 (9.7)
Nurse	34 (32.4)	1 (7.7)	11 (35.5)
Personal trainer	3 (2.9)	1 (7.7)	1 (3.2)
Physiotherapist	10 (9.5)	0 (0)	3 (9.7)
Social worker	2 (1.9)	1 (7.7)	1 (3.2)
Specialist	22 (21.0)	0 (0)	8 (25.8)
None	34 (32.4)	1 (7.7)	6 (19.4)

^^^ Participants may have selected multiple sources of PA education. * Pearson Chi-Square *p* < 0.05. ^c^ OR = 1.43, 95% CI: 0.25, 2.79, *p* < 0.05.

## Data Availability

The data presented in this study are available on request from the corresponding author due to ethic requirements.

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
