# Peer review of "Exploring Physical Activity Levels, Barriers, and Education Sources in People with Cancer Undergoing Chemotherapy"

_cancers, 2025, doi:10.3390/cancers17182987_

Round 1
Reviewer 1 Report
Comments and Suggestions for Authors
Add caption to the figure in page 2 and cite it in the text.
The introduction part should be updated to include 2025.
The results part should be merged with the discussion part under the title “Results and discussion”.
The Citations in the discussion part should be transferred to the introduction part as discussion should only show the authors point of view and their own interpretation of the results.
Author Response
We thank you for the feedback on this manuscript. Please see attachment with the authors comments and changes made.

Reviewer 2 Report
Comments and Suggestions for Authors
This is a study presenting results of a survey of outpatients with cancers receiving chemotherapy, to explore perceived barriers to physical activities and to identify gaps in services. The authors identified that few patients meet PA recommendations during treatment, and highlighted gaps in patient care and education, similar to previous studies on this subject which have been conducted, so it's not particularly novel although it is relevant in that it has identified that additional resources are required to improve access for patients.
It would perhaps be more useful if the survey is conducted over a longer period, in the more recent period (the survey was distributed in 2018?) and across specific cancer groups (haematological vs oncological) as there are very limited conclusions that can be drawn based on the current numbers and the very heterogenous patient groups (different schedules of treatment with different side effect profile). It would also be useful to capture pre-treatment or pre-diagnosis of cancer physical activity level, and if this has changed with treatment and also after treatment with resolution of side effects, in terms of trying to establish if there would be quantifiable benefits. As increasingly recognized, there are problems with obesity in the western society, and it is difficult to establish the current level of physical activity in people without cancer. I appreciate that this is a difficult and complex area to get data, as patients with cancers have different needs - dietary, psychological, financial and addressing the aspect of physical activities is challenging with these all having the potential to impact on the level of physical activity. Whilst the validated questionnaire is a reasonable tool to use in this study, as pointed out by the authors, it only captures patients who agree to participate (biased response) and likely excluded many who may have contributed additional useful insights.
Overall, this study does provide useful insights into low PA levels in patients with cancer having chemotherapy, and highlighted areas for future research, but is limited in its ability to draw significant conclusions from the derived data.
Author Response
Thank you for your time and detailed responses. Please see attached responses and changes based on your feedback

Reviewer 3 Report
Comments and Suggestions for Authors
Why did you not go on to increase the sample size to the needed number? You should give at least give a short explanation, because htis realy the big limitation of your trial.
Author Response
We would like to thank reviewer 3 for the time and effort they put into reviewing this paper. Please see the attachment with responses and changes made as a result of your feedback.

Round 2
Reviewer 1 Report
Comments and Suggestions for Authors
The authors responded adequately to the comments and the manuscript is now suitable for publication in the current revised form.
Author Response
Thank you for your time and input.
Reviewer 2 Report
Comments and Suggestions for Authors
Thank you for your responses, which addressed most of my comments. I note the additions of details on the heterogeneity of the data, the length of the recruitment period into the manuscript, these sentences added to its clarity and purpose, thank you. Although of note the survey was conducted 7 years ago so more recent, larger studies, would be beneficial to increase our understanding of this topic and to make it more relevant.
Author Response
Thank you for your time to review the changes made to the manuscript
Comments 1: Although of note the survey was conducted 7 years ago so more recent, larger studies, would be beneficial to increase our understanding of this topic and to make it more relevant
Response 1:
Despite the study being completed 7 years ago, it is relevant to the current status of exercise oncology and is supported by more recent literature in the discussion. We have also added additional support for this to the discussion.
Additional reference:
[35] Chan, A.; Ports, K.; Neo, P.; Ramalingam, M.B.; Lim, A.T.; Tan, B.; Hart, N.H.; Chan, R.J.; Loh, K. Barriers and facilitators to exercise among adult cancer survivors in Singapore. Support Care Cancer 2022, 30, 4867-4878, doi:10.1007/s00520-022-06893-y.
Page 10, Line 282-283: (Existing text with additional recent reference added for support). "These findings align with existing literature, which consistently demonstrates low PA adherence across the cancer continuum[16,34,35]."
Page 12, Line 374-376: "A recent study in people undergoing chemotherapy indicated similar trends with education, whereby 73% of participants received guidance about PA during treatment[35]."
We acknowledge that the survey was completed seven years ago now and that larger studies would be beneficial to increase understanding of the topic. We have attempted to make this clearer in the future research section of the discussion.
Page 14, Line 465-468 - "Finally, studies with larger sample sizes that aim to identify changes in physical activity, ranging from pre-diagnosis and pre-chemotherapy to post-treatment would be beneficial to gaining a full understanding of the impact that chemotherapy has on PA levels.
Comment 2: Conclusions supported by the results: Can be improved.
Response 2: Thank you for your feedback on the conclusion. We feel that this study does highlight the critical gap in care in Australia, with no recent literature disputing this. The conclusion identifies the need for exercise specialists to be integrated into standard care, aligning with recommendations for cancer care with are referenced in the manuscript discussion. We do acknowledge your feedback that the study was completed 7 years ago, and have therefore changed the conclusion to remove the "urgent need" for change. The updated conclusion is below.
Page 14, Line 471-480 -
"This study reveals a critical gap in PA levels among individuals undergoing chemotherapy in Australia, with a substantial proportion of people engaging in no MVPA or resistance exercise. Alarmingly, nearly one-third of participants in this study reportedly received no PA education during their treatment. It highlights the need for improved services to promote PA and integrate exercise specialists, such as AEPs into standard cancer care. All individuals undergoing chemotherapy can benefit from being physically active, and it is essential that they are offered this opportunity. Consistent and reliable individualised PA support is crucial for empowering patients to engage in safe and effective PA during their treatment, ultimately improving their quality of life and overall health outcomes."
Thank you again for your time and input to ensure the impacts of this research are clear and a useful addition to the current literature.